# Supersulfated Cement Applied to Produce Lightweight Concrete

**DOI:** 10.3390/ma14020403

**Published:** 2021-01-15

**Authors:** Liliya F. Kazanskaya, Olga M. Smirnova, Ángel Palomo, Ignacio Menendez Pidal, Manuel Romana

**Affiliations:** 1Department of Building Materials and Technology, Emperor Alexander I St. Petersburg State Transport University, 190031 Saint-Petersburg, Russia; yalifa@inbox.ru; 2Department of Constructing Mining Enterprises and Underground Structures, Saint-Petersburg Mining University, 199106 Saint-Petersburg, Russia; smirnovaolgam@rambler.ru; 3Department of Materials, Eduardo Torroja Institute for Construction Science, 28033 Madrid, Spain; palomo@ietcc.csic.es; 4Departamento de Ingeniería y Morfología del Terreno, Universidad Politécnica de Madrid, 28040 Madrid, Spain; manuel.romana@upm.es

**Keywords:** phosphogypsum, expanded-clay aggregates, coefficient of thermal conductivity, frost resistance, structure formation, circular economy

## Abstract

The physical and mechanical characteristics of expanded-clay lightweight concrete based on a supersulfated binder in comparison with lightweight concrete based on ordinary Portland cement were studied. In replacing CEM 32.5 with a supersulfated binder of 6000 cm^2^/g specific surface, one can increase the tensile strength in bending up to 20% and can increase the ratio of the tensile strength in bending to the compressive strength that indicates the crack resistance increase of concrete. Compressive strengths at the age of 28 days were equal to 17.0 MPa and 16.6 MPa for the supersulfated binder of 3500 cm^2^/g specific surface and CEM 32.5, respectively. Shrinkage deformation of hardening concrete, indicators of fracture toughness, frost resistance, and thermal conductivity were determined during the experimental works. The coefficient of thermal conductivity decreased up to 12% compared to the use of CEM 32.5. An enhancement in concrete properties was associated with the increase of supersulfated binder fineness.

## 1. Introduction

The issue of environmental protection is becoming global with active human intervention in natural resources. Two equally important problems should be highlighted: first, protection from depletion as well as reproduction of natural resources and, second, protection from pollution of the environment due to production processes associated with the extraction, transport, storage, and processing of raw materials as well as with the formation of harmful waste [1,2].

Trends in the development of industrial production and the construction industry in particular provide the widespread use of secondary raw materials. Multi-tonnage wastes from the chemical and metallurgical industries are important raw materials for construction [3,4,5].

Modern methods of utilization of such wastes and by-products confirm the possibility of their use in various directions in construction, in particular, for producing low-energy non-burning binders such as supersulfated slag binders [6,7]. Thus, three main problems can be solved:-making the production of binders based on secondary resources cheaper than the production of binders from natural raw materials [8,9];-ensuring environmental cleanliness and waste-free production [10];-obtaining binders with sufficiently high quality and with stability of the main characteristics [11,12,13].

Supersulfated cements consist of ground granulated blast furnace slag, gypsum, and a hardening activator. Lime or Portland cement are traditionally used as slag hardening activators in so-called alkaline activation. Natural gypsum is also used as a sulfate component, i.e., combined (sulfate) activation [14]. A supersulfated binder (SSB) compared to ordinary Portland cement (OPC) has advantages such as low heat release during hydration, resistance to sulfates contained in the sea or ground water [15,16]. The main research in recent years has been devoted to the development of eco-friendly SSBs based on industrial by-products and wastes: slags, ash, phosphogypsum, cement dust, or lime-containing by-products of the chemical industry [16]. This makes it possible to completely eliminate the use of OPC in the composition of these binders with comparable strength characteristics to that of OPC. This technology is a model of circular economy when it is profitable to reuse resources closing the chain: resources–goods–wastes–resources [17,18]. In this case, the production becomes self-sufficient and there is no need to search for new resources.

Structure formation, namely ettringite formation, is of great importance in the hardening of SSBs. Accordingly, it is necessary to select a binder composition to exclude ettringite formation in the hardened matrix that can lead to strength decreases [19,20].

A favorable course of dissolution processes for the binder will be provided when sufficiently intensive dissolution of the slag alumina is realized under conditions of high saturation of the solution with SO_4_^2−^ ions. Under these conditions, the high-sulfate form of calcium hydrosulfoaluminate ettringite (C3A·3CS·31H) is formed by the through-solution mechanism with the formation and development of crystallohydrates. The crystallohydrates are placed in the intergranular space, compacted, and strengthens the cement matrix in these conditions without expansion and destruction [20].

It is important that the formation of crystalline high-base calcium hydrosulfoaluminates ends in the initial period of hardening. During this period, the matrix still has plastic properties and volume changes that occur during crystallization do not disrupt the matrix structure [21,22,23].

Stresses in the solidified structure are especially dangerous when the initial substances in the aqueous solution react with the components of the solid phase of the binder. In this case, the particles of newly formed compounds lead to a disturbance of the matrix structure [24,25,26].

The hydration products do not cause dangerous stress values and contribute to matrix hardening when such reactions occur in the liquid phase. The nature of the reaction flow depends on the amount of dissolved alumina, which is the function of the lime concentration in the liquid system of the slag—calcium sulfate—water [20,27,28,29,30].

It is known that regulated structure formation of hardening cement pastes results in an increase in concrete durability [31,32,33]. Control of matrix structure formation allows for increasing the physical and mechanical characteristics of lightweight concrete based on OPC or on alkali-activated binders [34,35,36,37,38]. The enhancement of binder fineness increases its hydration degree, changes the matrix structure, and improves the properties of lightweight concrete based on OPC [39,40,41,42,43]. However, the number of published results concerning the effect of composition and fineness of SSBs on the properties of expanded-clay concrete is not enough.

The aim of the paper is theoretical and experimental justification of technological approaches for improving the concrete properties of lightweight concrete based on expanded-clay gravel and a supersulfated binder.

## 2. Significance of Research

The developed technology is a model of circular economy when it is profitable to reuse resources closing the chain: resources–goods–waste–resources. The improvement of mechanical and physical properties of lightweight concrete was obtained by increasing the fineness of supersulfated cement based on secondary resources that contributed to the increase of the adhesive strength of the contact zone between expanded clay and the matrix.

## 3. Experimental Methodology

### 3.1. Materials

The supersulfated binder was obtained by mixing the ground granulated blast furnace slag (ggbfs) from Public Joint Stock Company (Magnitogorsk, Russia) “Magnitogorsk iron and steel works”, the sulfate component—phosphogypsum (fg)—from JSC (Sterlitamak, Russia) “Meleuzovskie mineral fertilizers” as well as cement kiln dust (ckd) from JSC “Bashkir soda company”. The cement kiln dust was used as a hardening activator. Chemical compositions of components have been confirmed by manufacturer’s product bulletins provided by manufacturers. Characteristics of the slag are given in Table 1 and Table 2. The chemical compositions were obtained according to [44].

The hydraulic properties of blast-furnace granulated slag were evaluated using the K quality coefficient according to [45]. This coefficient is calculated by a formula which, when the content of magnesium oxide MgO is up to 10%, takes the following form:$$K=\frac{\%\mathrm{CaO}+{\%\mathrm{Al}}_{2}O_{3}+\%\mathrm{MgO}}{{\%\mathrm{SiO}}_{2}+{\%\mathrm{TiO}}_{2}}$$

The basicity modulus of slag is the ratio of the amount of alkaline oxides CaO + MgO to the sum of the oxides of silicium and aluminum SiO_2_ + Al_2_O_3_. The activity modulus of slag is the ratio of the amount of Al_2_O_3_ to SiO_2_.

The preliminary study of wastes from the chemical industry in the Southern Urals region in Russia showed the presence of a number of components that can be used as lime- and sulfate-containing hardening activators for clinker-free binders without significant refinement [20]. These wastes can be used instead of expensive artificially synthesized components without significantly reducing the technical properties of concrete.

The cement kiln dust has 6–7% of free lime and up to 1.5% of K_2_O + Na_2_O. The chemical composition of the cement kiln dust is shown in Table 3.

Phosphogypsum is the waste of sulfuric acid treatment of apatite and phosphorite concentrates obtained in the process of orthophosphoric acid production. The most common process of dihydrate decomposition of calcium–fluorapatite in industrial conditions is carried out at the temperature 65–80 °C. The resulting product contains phosphoric anhydrite in a quantity of 28–32% and sludge of CaSO_4_·2H_2_O. The yield of dry phosphogypsum waste is 4–5 t per 1 t of P_2_O_5_ since the ratio of mass fractions of CaO/P_2_O_5_ in the phosphate raw material varies from 1.35 to 1.65.

The phosphogypsum from JSC “Meleuzovskie mineral fertilizers” used in these experiments as a sulfate-containing activator is characterized by the following composition: CaSO_4_·2H_2_O—94.6%, CaHPO_4_—traces, Ca_3_PO_4_—1.31%, and H_3_PO_4_ and Ca(H_3_PO_4_)_2_—0.47% as shown in Table 4.

### 3.2. Methods

#### 3.2.1. Design of Lightweight Concrete Composition

The maximum content of alkaline additives was assigned to ensure intensive dissolution of slag aluminates and uniform change of the binder volume at hardening, which is achieved by limiting the amount of calcium oxide, CaO_free_, up to 2% of the binder mass. According to research in [20], the concentration of the sulfate component in the compositions of complex-activated slag binders is determined by the fact that the amount of calcium sulfate should be sufficient for maximum binding of aluminates and should not be greater since this will cause an increase in the inert phase amount in the binder and will reduce its activity.

Expanded-clay coarse and fine aggregates have been used in the lightweight concrete compositions according to the data in Table 5.

The water demand of lightweight aggregates was defined as absorption at mixing to ensure a water–binder ratio. Water absorption is determined in percent as the difference in the mass of the aggregate before and after saturation with water. The aggregate is placed in a container with holes and a lid. The container is slowly submerged in water and shaken to remove air bubbles from the aggregate. The container with the aggregate is kept in water for 1 h, then removed from the water, and suspended so that the excess water is draining for 10 min. Then, the aggregate mass is determined. The average value of the results of two parallel tests is taken as the test result.

Experimental binder compositions were prepared with separate grinding of slags and activators followed by mixing the dry ingredients in the laboratory drum and ball-type mill with the diameter of the balls at 30 mm. The fineness of component grinding was chosen to achieve both a high activity of the binder and economic feasibility. The specific surface was determined using the Blaine method.

The use of dump phosphogypsum with a natural moisture content of about 25% and specific surface area of 3000 cm^2^/g as a sulfate component of the binder gives a lower matrix strength especially in the early stages and after heat-steaming treatment. Repulping and heat treatment of phosphogypsum make it possible to obtain the binder with higher activity. The sample strength in this case exceeds the sample strength of the SSB with natural sulfate components.

Samples of lightweight concrete based on SSBs with S_sp_ = 3500 cm^2^/g and S_sp_ = 6000 cm^2^/g were made at water-to-binder ratios equal to W/B = 0.55 and W/B = 0.50 for the B15 and B25 strength classes, respectively. The slump was 5–7 cm. The compositions of concrete are presented in Table 6.

Lightweight concrete composition can be calculated by different methods. Three series of test mixes were prepared to determine the optimal composition of the required grade according to the average density and compressive strength class. The following recommendations were used for choosing the composition. The recommended amount of cement of CEM 32.5 strength class was 300–400 kg/m^3^ for the B25 concrete strength class. Water was assigned in the amount of 10–15% of the mass of the dry components depending on the aggregate porosity and the required workability. The total amount of aggregates per 1 m^3^ can be determined by the following formula:A = D − 1.15 × C
where

A—the total amount of coarse and fine aggregates, kg/m^3^

D—the required grade of the average density of concrete, kg/m^3^

C—the cement amount, kg/m^3^

The coefficient 1.15 takes into account chemically bound water that is involved in cement hydration. The recommended volume of expanded clay gravel per 1 m^3^ of concrete with the 5–7 cm slump is 0.85–0.9 m^3^. The increase of the volume of expanded-clay gravel per 1 m^3^ leads to strength decrease. The mass of the coarse and fine aggregates can be defined by taking into account the bulk density of coarse aggregate (ρ_b_):A_corse_ = (0.85 ÷ 0.9) × ρ_b_, kg/m^3^
A_fine_ = A − A_corse_, kg/m^3^

Test mixes were performed, and the water amount was corrected to obtain the required workability. The average density of fresh concrete mix (ρ) was checked after compaction, and the actual material consumption per 1 m^3^ was determined by the following formula:M = ρ × m/∑
where

M—amount of a component in 1 m^3^, kg

m—amount of the same component in bath, kg

∑—sum of all components in bath (cement, coarse and fine aggregates, water), kg

The graph of the strength dependence on the average density and on the cement amount was performed on the data of three series of test mixes and the optimal composition was selected from the graph.

Calculation example for concrete of D1500 and B25:

Three cement amounts were assigned based on the recommendations: 360, 380, and 400 kg/m^3^. Then, the composition was calculated for each cement amount, for example, for 380 kg/m^3^:A = 1500 − 1.15 × 380 = 1063 kg/m^3^
A_corse_ = 0.85 × 790 = 671.5 kg/m^3^
A_fine_ = A − A_corse_ = 1063 − 671.5 = 391.5 kg/m^3^
W = 13%(380 + 671.5 + 391.5) = 188 kg/m^3^

Calculation example for concrete of D1300 and B15:

Three cement amounts were assigned based on the recommendations: 260, 280, and 300 kg/m^3^. Then, the composition was calculated for each cement amount, for example, for 280 kg/m^3^:A = 1300 − 1.15 × 280 = 978 kg/m^3^
A_corse_ = 0.9×567 = 510 kg/m^3^
A_fine_ = A − A_corse_ = 978 − 510 = 468 kg/m^3^
W = 12%(280 + 510 + 468) = 151 kg/m^3^

#### 3.2.2. Testing the Lightweight Concrete

The concrete frost resistance was determined according to [46]. The coefficient of water resistance was defined as the ratio of compressive strength at 28 days when samples were stored in water to the compressive strength when samples were stored in air. Compressive strength and flexural strength were determined according [47,48] using the TONI TECHNIK universal test station.

The elasticity modulus test was done by stage loading of prism specimens of 100 × 100 × 400 mm^3^ size with compressive axial load until failure to determine the prism strength. Loading was done at the level of 30% of the failure load by measuring the sample deformation to determine the elasticity modulus. The sensors were attached to the sample. The measurement accuracy of the sensors was 0.001 mm. The sensor base for determination the longitudinal deformation was 150 mm. The prismatic strength R_pr_ was calculated for each sample by the following formula:R_pr_ = P_p_/F
where

P_p_—is the failure load measured on the press scale;

F—is the average value of the cross-sectional area of sample.

The elasticity modulus E_δ_ was calculated for each sample at the load level of 30% of the failure load according to the following formula:E_δ_ = Ϭ_1_/ε_1_
where Ϭ_1_ = P_1_/F is the stress increment from conditional zero to the level of the external load equal to 30% of the failure load;

P_1_—is the external load increment; and

ε_1_—is the increment of the elastic-instantaneous relative longitudinal deformation of sample at the load level equal to P_1_ = 0.3P_p_ and measured at the beginning of each stage of loading.

Several steady and transient methods can be used to measure the thermal conductivity of building materials [49,50]. In this study, the MIT-1 device for defining the thermal conductivity of concrete was used in accordance with the standard 30256-94 “Building materials and products. Method of thermal conductivity determination with cylindrical probe”. The method for determining the thermal conductivity with a cylindrical probe under a non-stationary thermal regime in the temperature range of 90–573 K is based on the dependence of the temperature of the heated body (cylindrical probe) embedded in the material on thermal conductivity of the material surrounding the probe. The MIT-1 device consists of an electronic unit, a heat probe, and a power supply unit, which is necessary to ensure sufficient power of the heater during the measurement process (Figure 1). The principle of operation of the device is based on measuring the temperature change of a probe over a certain time when it is heated at constant power. A material sample must be prepared as follows before measurement: A hole is drilled in the sample. The probe with a diameter of 3 mm is firmly inserted into the prepared hole. The gap between the wall of the hole and the probe inserted into it should be no more than 0.1 mm. Before the first measurement, it is necessary to keep the sample material and the probe at measurement temperature for at least two hours.

For shrinkage test, the specimens were three prisms of 40 × 40 × 160 mm^3^ size for each composition according to [51]. The device for determining the shrinkage deformations of samples is shown in Figure 2. The prepared sample should be installed into a device for testing and the initial readings should be taken from the indications of devices for measuring shrinkage deformations. The relative values of deformations at different ages were calculated from the absolute values of deformation using the formula:ε = ∆*l*/*l*
where *l*—base for measuring deformations, mm.

## 4. Experimental Results and Discussion

The presence of fine aggregates of expanded clay contributes to increased strength in the case of a binder with higher specific surfaces, as shown in Table 7. The average values obtained by statistical processing of 25 series of concrete samples for each strength class are given. The coefficient of variation was in the range of 4.5–6.5%. A series of control samples was three samples made from a single batch of concrete mix, hardened under the same conditions, and tested at the same age to determine the actual strength.

An increase of the ratio of tension strength in bending (R_tb_) to compressive strength (R_comp_) was characteristic of defects reduction and structure improvement [52,53].

Properties of lightweight concrete on porous aggregates depend on the state of contact zone between expanded the clay aggregate and cement matrix [17,54,55,56,57], and the density of the cement matrix [57,58].

Improving the structure of the expanded clay concrete based on SSB is caused by changing the conditions of structure formation: the decrease of the paste viscosity due to the increased specific surface area leads to a reduction of the viscosity of fresh concrete and provides better clogging of the open pores of expanded-clay aggregates. Under these conditions, the contact surface of the cement matrix with aggregates increases, which contributes to the intensification of contact interaction processes.

The reactivity of SSB, namely, the binding process of free lime and gypsum with the aluminate phases of slag and aluminosilicate phases of an expanded-clay aggregate depends on binder composition. The contact zone is formed more intensively due to hydration reactions of SSB with the expanded-clay aggregate. These hydration products provide the growth of adhesive strength between aggregates and the cement matrix. Fine expanded-clay fractions also provide the growth of hydration products in the contact zone, ensuring the homogeneity of the pore structure of a cement matrix.

The pore structure changes can be explained by the binder hydration acceleration and more intense interaction of phosphogypsum and amorphous phases of slag with ettringite formation, where ettringite plays the reinforcing role. X-ray phase analysis indicates the complete binding of Ca(OH)_2_ in concrete based on a binder with greater specific surface [20]. There are no lines of diffraction reflections of Ca(OH)_2_ in the cement matrix. This concrete structure formation leads to increasing strength and deformation characteristics, as shown in Table 7, Table 8 and Table 9.

The increase in the R_tb_/R_comp_ ratio corresponds to defect reduction in the concrete structure. In replacing Portland cement CEM 32.5 with SSB of higher specific surface, one can increase the R_tb_ value up to 20% and can increase the ratio of R_tb_/R_comp_ as shown in Table 7, which indicates the crack resistance increase of concrete.

The increase of deformability of a lightweight concrete based on SSB is caused by changing the pore structure of concrete, namely, the uniform distribution of conditionally closed pores that act as dampers in this case.

The compressive strength at the ages 3–24 months, the coefficient of water resistance, and frost resistance of lightweight concrete on SSB are not lower than the same characteristics of lightweight concrete on OPC according to Table 8 and Figure 3. Frost resistance of concrete based on SSB is close to the values using CEM 32.5 in expanded-clay concrete.

The uniform pore distribution in concrete structure improves its deformative properties and increases the structure resistance to internal pressures. It was stated that the elasticity modulus (E_δ_) of studied lightweight concrete is slightly higher than the elasticity modulus of lightweight concrete based on CEM 32.5. The ratio R_tb_/E_δ_ indirectly characterizes the fracture toughness of concrete. The ratio R_tb_/E_δ_ increases by 1.14–1.18 times when using the SSB, as shown in Table 9.

The presence of low-modulus additives or hydration products in a concrete structure provides an increase in crack resistance and frost resistance of concrete since low-modulus additives can be energy crack dampers [59,60].

The required density of lightweight concrete is obtained by filling the concrete volume with fine and coarse expanded-clay aggregates. A satisfactory performance of the strength and physical properties of concrete is due to the use of SSB including its finer grinding with optimization of the pore structure. This leads to the improvement of thermal and physical characteristics for lightweight concrete. The coefficient of thermal conductivity decreased up to 12%, as shown in Table 10.

The changes in mass and size of expanded-clay concrete prisms based on SSB and OPC are shown in Figure 4. The latter shows that samples of concrete on OPC have much greater shrinkage than samples based on SSB for long-term storage in air.

The most significant shrinkage deformations of concrete on OPC are observed during the first 5 days when samples intensively lose water. During further air storage, the shrinkage deformation was attenuated and, after 60 days, reached values of 0.55 mm/m. Samples based on SSB by the end of the second month had only shrinkage equal to 0.04 mm/m.

Ettringite is one of the main hydration products of the supersulfated binder [6,11,13,20]. Its volume exceeds the volume of the initial substances involved in the formation of ettringite. It is necessary to select a binder composition to exclude ettringite formation in the hardened matrix that can lead to structure destruction. Ettringite formation must end in the initial period of hardening. During this period, the matrix still has plastic properties and volume changes that occur during crystallization do not disrupt the matrix structure [21,22,23]. However, it is necessary to choose a supersulfated binder composition for which expansion from the ettringite crystallization is sufficient to compensate the reduction of concrete volume due to shrinkage.

## 5. Conclusions

The supersulfated binder containing phosphogypsum equal to 20% of the slag mass and cement kiln dust equal to 10% of the slag + phosphogypsum mass ensures the formation of durable structure of lightweight concrete using energy- and resource-saving technologies.

The improvement of durability characteristics of concrete was proven experimentally. Increasing the specific surface area of the supersulfated binder from 3500 to 6000 cm^2^/g leads to improvements in the physical and mechanical characteristics of lightweight expanded-clay concrete in comparison with lightweight expanded-clay concrete on ordinary Portland cement: compressive strength at 28 days equal to 18.2 MPa and 16.6 MPa for supersulfated binder and CEM 32.5, respectively. In replacing CEM 32.5 with a supersulfated binder of higher specific surface, one can increase the tensile strength in bending up to 20% and can increase the ratio of the tensile strength in bending (Rtb) to the compressive strength, which indicates the crack resistance increase of concrete.

The most significant shrinkage deformations, 0.36 mm/m, of concrete on CEM 32.5 were observed during the first 5 days when samples intensively lose water. During further air storage, shrinkage deformation was attenuated and, after two months, reached values of 0.55 mm/m. Samples based on the supersulfated binder had shrinkage equal to 0.01 mm/m after first 5 days and shrinkage equal to 0.04 mm/m by the end of the second month. The compressive strength at the age of 3–24 months, coefficient of water resistance, and frost resistance of lightweight concrete on the supersulfated binder were not lower than the same characteristics of lightweight concrete on CEM 32.5. F50 and F75 frost resistances were found for the B15 and B25 concrete strength classes, respectively. It was stated that the elasticity modulus (E) of the studied lightweight concrete is slightly higher than the elasticity modulus of lightweight concrete based on CEM 32.5. The ratio Rtb/E indirectly characterizes the fracture toughness of concrete. The ratio Rtb/E increases by 1.14–1.18 times when using the supersulfated binder. The coefficient of thermal conductivity decreased up to 12% compared to the use of CEM 32.5.

## Figures and Tables

**Figure 1 materials-14-00403-f001:**
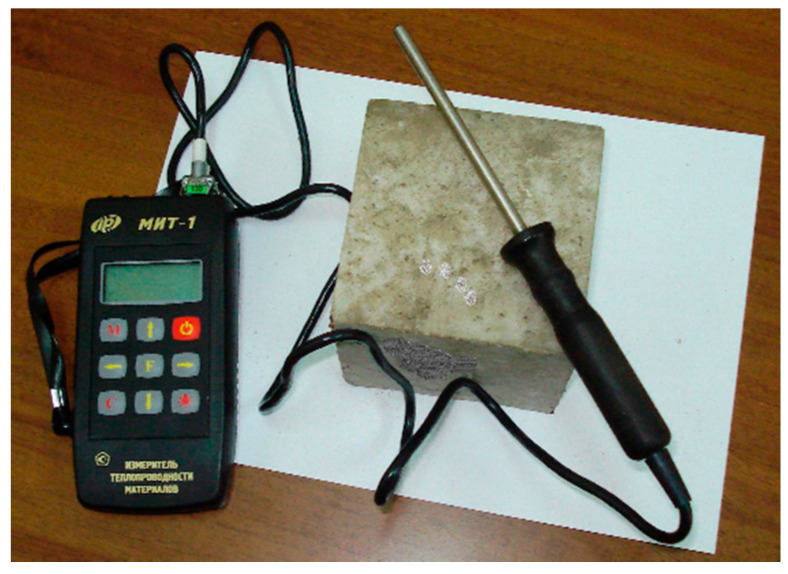
Device for determining the thermal conductivity.

**Figure 2 materials-14-00403-f002:**
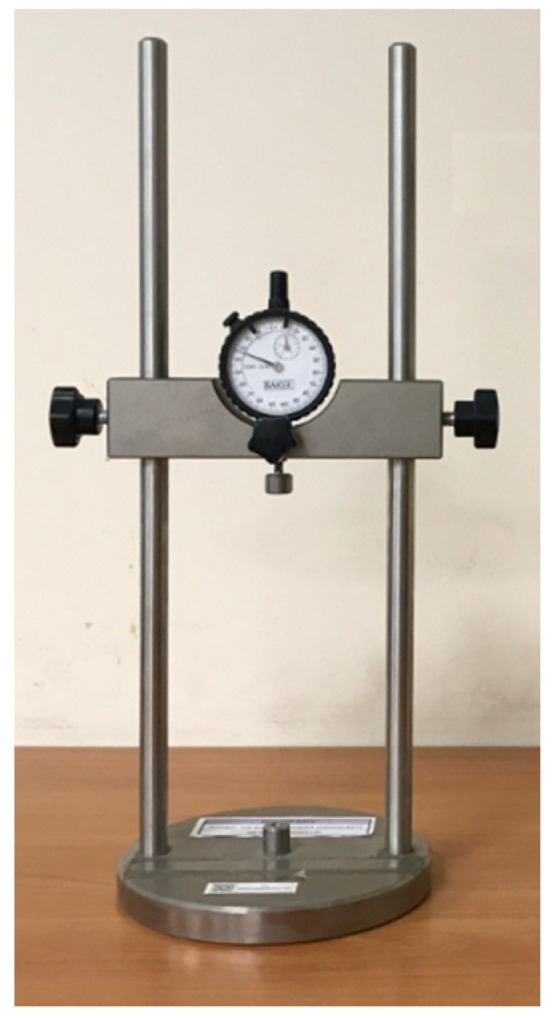
Device for determining the shrinkage deformation.

**Figure 3 materials-14-00403-f003:**
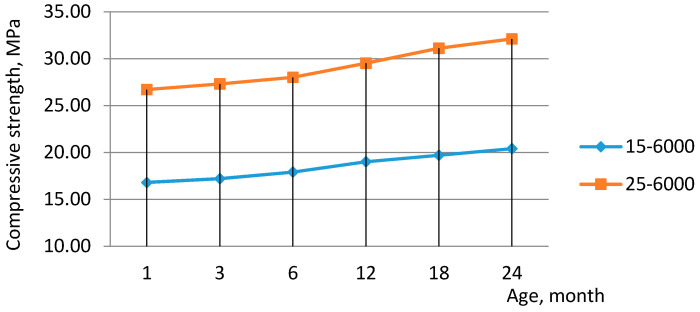
Evolution of compressive strength in the period of 1–24 months.

**Figure 4 materials-14-00403-f004:**
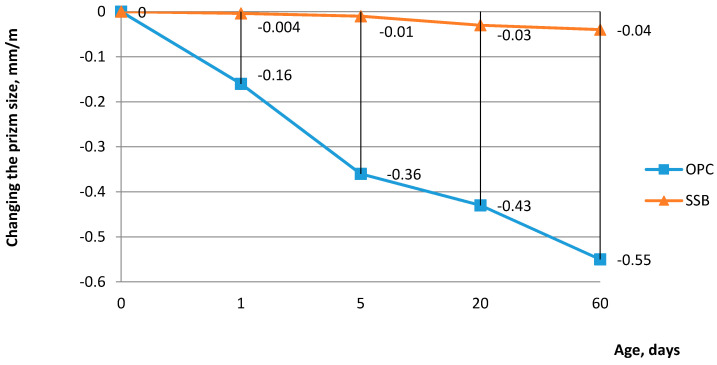
Shrinkage strain for long-term storage in air: SSB—supersulfated binder (composition 15-6000); OPC—Ordinary Portland cement (composition 15-cem).

**Table 1 materials-14-00403-t001:** Chemical composition of slag, %.

SiO_2_	A1_2_O_3_	CaO	MgO	SO_3_	FeO	Fe_2_O_3_	MnO	TiO_2_	LOI
35.68	14.32	40.09	5.54	1.17	—	0.84	0.80	0.80	1.56

**Table 2 materials-14-00403-t002:** Modular characteristics of slag.

Basicity Modulus	Activity Modulus	Quality Coefficient
0.90–1.05	0.40–0.45	1.66

**Table 3 materials-14-00403-t003:** Chemical composition of cement kiln dust, %.

SiO_2_	CaO	MgO	K_2_O + Na_2_O	A1_2_O_3_	Fe_2_O_3_	SO_3_	(CaO + MgO)_free_	Cr
13–15	42–45	2–3	1.0–1.5	3–6	2.5–4.0	0.7	6–7	0.3–0.5

**Table 4 materials-14-00403-t004:** Chemical composition of phosphogypsum, %.

CaO	SO_3_	H_2_C	P_2_O_5_	R_2_O_3_	R_2_O	LOI
31.5	46.76	18.9	0.89	0.24	0.42	1.29

**Table 5 materials-14-00403-t005:** Characterization of lightweight concrete components.

Component	Characteristics
Expanded clay coarse aggregate for concrete of D1300 and B15	Water demand—10.8% Bulk density—567 kg/m^3^ Strength when compressed in the cylinder—2.3 MPa Grain composition of expanded clay: 5–10 mm—40% 10–20 mm—60%
Expanded clay coarse aggregate for concrete of D1500 and B25	Water demand—9.6% Bulk density—790 kg/m^3^ Strength when compressed in the cylinder—2.9 MPa Grain composition of expanded clay: 5–10 mm—40% 10–20 mm—60%
Expanded clay fine aggregate	Water demand—9.4% Bulk density—800 kg/m^3^ Fineness modulus = 1.8
Supersulfated binder	Type 1. Composition *: 72%ggbfs + 18%fg + 10%ckd Specific surface (S_sp_)—3500 cm^2^/g Standard consistency by Vicat method—30% Setting time: begin—2 h 40 min; end—7 h Compressive strength at the age of 28 days, R_comp_ = 40 MPa Type 2. Composition: 72%ggbfs + 18%fg + 10%ckd Specific surface (S_sp_)—6000 cm^2^/g Standard consistency by Vicat method—32% Setting time: begin—2 h 20 min; end—6 h Compressive strength at the age of 28 days, R_comp_ = 43 MPa
Mixing water	Drinking tap water

*—the ratio of binder components is based on the previously performed series of experiments [13,16].

**Table 6 materials-14-00403-t006:** Compositions of lightweight concrete.

Concrete Strength Class	B15			B25		
Concrete Average Density Class	D1300			D1500		
Designation	15-3500	15-6000	15-cem	25-3500	25-6000	25-cem
Type of binder	SSB-3500	SSB-6000	CEM 32.5	SSB-3500	SSB-6000	CEM 32.5
Binder, kg/m^3^	280	280	280	380	380	380
Fine aggregate, kg/m^3^	460	460	460	390	390	390
Coarse aggregate, kg/m^3^	510	510	510	670	670	670
Water, kg/m^3^	154	154	154	190	190	190

Note: SSB-3500—the concrete with binder of S_sp_ = 3500 cm^2^/g, SSB-6000—the concrete with binder of S_sp_ = 6000 cm^2^/g.

**Table 7 materials-14-00403-t007:** Mechanical properties of lightweight concrete.

Concrete Strength Class		B15			B25		
Concrete Average Density Class		D1300			D1500		
Binder		SSB-6000	SSB-3500	CEM 32.5	SSB-6000	SSB-3500	CEM 32.5
Strength at 28 days, MPa	Tensile strength in bending, R_tb_	5.1	4.9	4.2	7.3	6.9	6.1
	Prism compressive strength, R_pr_	17.2	16.1	15.8	27.4	26.5	25.4
	Cubic compressive strength, R_comp_	18.2	17.0	16.6	29.0	28.2	27.0
$\frac{R_{\mathrm{tb}}}{R_{\mathrm{comp}}}$		0.28	0.28	0.25	0.25	0.24	0.22

Note: SSB-3500—the concrete with binder of S_sp_ = 3500 cm^2^/g, SSB-6000—the concrete with binder of S_sp_ = 6000 cm^2^/g.

**Table 8 materials-14-00403-t008:** Durability characteristics of lightweight concrete.

Designation	Compressive Strength, MPa				Frost Resistance	Coefficient of Water Resistance
	After Freeze-Thaw Cycles					
	50	75	100	150		
15-6000	16.9/100.3 *	17.7/105 *	17.3/103 *	16.8/100.2 *	F50/F50 **	0.93/0.89 **
25-6000	26.6/101.5	28/107 *	27.5/105 *	26.5/101.1 *	F75/F75 **	0.95/0.9 **

Note: *—the values in % from strength at the age of 28 days; **—values for compositions of expanded clay concrete based on CEM 32.5.

**Table 9 materials-14-00403-t009:** Comparative study of strength and deformation characteristics of lightweight concrete on SSB and CEM 32.5.

Strength Average Density	R_tb_^1^ R_tb_^2^	Modulus of Elasticity E_δ_ ^1^.10^−2^, MPa			E_δ_^1^ E_δ_^2^		R_tb_^1^/E_δ_^1^ R_tb_^2^/E_δ_^2^
		0.2 R_pr_	0.3 R_pr_	0.4 R_pr_	0.2 R_pr_	0.5 R_pr_	
B 15 D1300	1.20	161	136	112.9	1.05	1.08	1.18
B 25 D1500	1.19	222	192.8	166.5	1.07	1.09	1.14

Note: Index 1 refers to the concrete based on the supersulfated binder with S_sp_ = 6000 cm^2^/g. Index 2 refers to the concrete based on CEM 32.5. Underlines correspond with the different values (up and down) shown in the table.

**Table 10 materials-14-00403-t010:** Heat conductivity of lightweight concrete.

Concrete	Average Density (kg/m^3^)					
	750	900	1050	1200	1350	1450
	Heat Conductivity W/(m·°K) for Conditions A/B					
15-6000	0.28/ 0.31	0.37/ 0.39	0.38/ 0.42	0.43/ 0.48	0.49/ 0.53	—
15-6000 (fine expanded-clay aggregate was replaced by quartz sand with the fineness modulus of 2.1)	—	0.35/ 0.39	0.41/ 0.45	0.46/ 0.51	0.54/ 0.58	0.65/ 0.69

Note: A—45% humidity condition; B—95% humidity condition.

## Data Availability

Data available in a publicly accessible repository.

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
