# Peer review of "Supersulfated Cement Applied to Produce Lightweight Concrete"

_materials, 2021, doi:10.3390/ma14020403_

Round 1

Reviewer 1 Report

All authors made great efforts to investigate the effect of supersulfated binder composition and fineness on the physical and mechanical characteristics of lightweight concrete. Most parts are clear. However, for better understanding of readers and to further clarify the present study, authors had better revise this paper with several comments below.

  1. In Table 5, The value of fineness modulus seems to be too low to be used in concrete mixture. Generally, the FM is 2.6 to 2.8 in concrete mix.
  2. In Table 5, the reviewer can not understand the composition of supersulphated binder. Total composition of supersulphated bind is not 100%. And what is the meaning of standard consistency?
  3. In Table 6, all author did not mention the slump or slump flow, as well as chemical admixture.
  4. In Table 6, what makes the big difference of mixing water of between B15 and 25 ?
  5. In Table 6, Why did all authors compare the CEM 32.5 with SBB, instead of CEM 42.5 ? In Table 5, The compressive strength of SBB at 28 days is more than 40 MPa.
  6. Some typos would be corrected.

        - Page 6, line 179, Rbt → Rtb

        - In Table 8, 05* → 105* (??)  

  1. In Figure 2, this result should be doublechecked with more samples and would be referred with some papers. The shrinkage test should be carried out for 3 months at least and the data reading should be measured every day at the early age.

Author Response

Please see attached file,

Thank you very much.

Authors, 

Reviewer 2 Report

I think this paper needs a lot work to be published. My comments to improve the manuscript are below.

The abstract should be revised. The first sentence is too long and confusing. More comprehensive results are expected in the abstract because is the most important for the potentials readers.

Introduction

I suggest to avoid nested references (i.e [1-4] in line 28 or [5-8] in line 31 that do not provide scientific interest or justification of the research carried out) and include a brief summary of the references.

There are several parts of the introduction section that are disconnected to others (ie lines 58-62). I think the introduction should be rewritten. The rationale should be highlighted. The statement of line 63 should  be referenced.

The authors mentioned in line 67-68 (… the use of SSB instead of OPC will exert some effect on the properties of the contact zone between the cement matrix and expanded clay gravel…). How can you justify this sentence with the research carried out? There are not SEM micrographs for analysing the ITZ zone between LWA and mortar matrix. Please revise the objectives and sentences written in the main text according to the results.

Experimental methodology

I suggest to use point and not comma for the decimal numbering. There is a mix in all the manuscript (see tables and main text).

I suggest to explain the results of Table 1 (FRX) and include the LOI value. I also suggest to explain Table 2. I do not understand the values given. Explain “modular characteristics”.

Can you explain the numbers given in Table 3? How can you explain 6…7%? I do not understand these values. I think that this table is obtained from a FRX analysis, please explain.

More detailed characteristics of lightweight aggregates is needed. What do you mean with water demand? Water absorption at 24 h, water absorption during mixing to maintain w/b ratio? Provide the commercial trade mark of the expanded clay.

I suggest to provide more detailed properties of setting time of supersulphated binder is needed.

Can you provide size distribution of the binder? This could be helpful.

Do you measure the consistency of the concrete after the manufacture?

Could you provide the standards used for all the tests carried out and their references? I think there is a mistake in standard EN 12390 (this is a group of standards, please provide their number).

Detailed information of thermal conductivity is needed.

Standard used for shrinkage test. The scheme of the test, is from the standard or it was done by yourself? Be careful with the copyrights.

Results and Discussion

From line 155 to line 188 is the initial paragraphs from this section. No results are shown and discussion is written. I suggest these paragraph in the introduction section.

Table 7. There are compressive strength in prisms and cubic specimen. I suggest to include in the methodology. How many samples? Can you provide the standard deviation or the coefficient of variation?

Table 8 is confusing. Can you provide a graph with the evolution of the compressive strength?

In table 9 there are values of modulus of elasticity and the methodology is not described. How?

MPA is MPa

Author Response

(The authors gave the same response as above.)

Reviewer 3 Report

1) Table 1 and table 4- Sum of the all components not give us 100%, Why?

2)Table 2 - what is mean 0,9...1,05 - this the range of? If yes please use "÷"

3) Please replace the sign - comma with a dot in numbers, we use the American system here

4) What method was used to design the concrete mix, because the authors do not say anything about it?

5) Please add the sieve curves of aggregate used in fabrication of concrete

6)Many authors are writing how to use a waste/raw materials for concrete production. Most of these manuscript are based on portland cement. When I browsing last Materials manuscripts the topic of concrete and portland cement are often discussed. To increase the rank of an article, it may be worth quoting authors from around the world, because most of your references is focusing on your domestic authors i.e. DOI:

-10.3390/ma13143189 or 10.3390/ma13214979

If you want to authors form other county you can use this one: 10.3390/ma13214989

This is only a suggestion, but these authors show how to make concrete mixes with waste materials, because topic of waste and concrete is very popular in last time. Please consider that. 

7)In my opinion Table 8 will be better visabe if you will prepare a graph of curves, because in this moment it difficult to say how the values are changing.

8) What about the statistic error calculations? Authors do not mean about it? The authors do not mention anything about it?

9) The table headings are not legible, please make another line with the measurement units for specific properties

10) all entries in references are not formatted as required. I am asking for improvement.

11) The division of labor by authors is also missing before references. Please add this paragraph

Author Response

(The authors gave the same response as above.)

Reviewer 4 Report

The article presents the study of the effect of the composition and fineness of the supersulfated binder on the physical and mechanical characteristics of lightweight concrete based on expanded clay aggregates compared to lightweight concrete based on ordinary Portland cement. This is a good quality article with a lot of results. This article may be published with minor corrections:

• Table 3, could you add the units.

Author Response

(The authors gave the same response as above.)

Round 2

Reviewer 1 Report

Thank you for your revision with your precious time.

But I do not agree to publish this papaer with this revision.

1) In Table 5, Contrary to what you answered, Fineness modulus of Expanded clay fine aggregate is still 1.8.

2) In Table 5, the type 1 and 2 composition of supersulphated binder are not revised as well.

3) In Table 6, If authors did not use a chemical admixture, how did you make the slump 5 to 7 cm?

    That is, even the water demand of aggregates is almost 20% (10.8%+9.8%), authors only used 154 kg/m3 for mixing water.   Please, provide some solid evidences on the paper.

4) As I know, the water absoption of artifical aggregate is a very important factor in lightweight concrete.

    Could you explain how to control it in detail ?

5) In line 281-288, if the description means the autogenous shrinkage because this chemical reaction occur within 1st month after concrete mixing, I would agree the test period.

    However, according to line 280, did you measure the shrinkage of SSB samples up to 2 years? why did not you mark on Figure 2?

Author Response

Dear reviewer, please find attached answer file.

Authors.

Reviewer 2 Report

I think the paper has been improved but it also needs more work to be published. I still think that a microstructural analysis is needed in this kind of work and in this type of journal. There are some gaps in the research that should be revised and included in the paper (more detailed information in the procedure with justification). Lightweight concrete is produced but it is not treated in the main text as clear as shown in the title. Water demand is not as clear as needed. The test and time should be stated to ensure the w/b ratio constant (effective). Shrinkage should be clearly plotted, with the common units. The shrinkage in early ages are more interesting and should be shown to evaluate the early behaviour only 3 measures in 60 days is insufficient.

Author Response

(The authors gave the same response as above.)

Reviewer 3 Report

Thank You very much for all answers. In this moment I recommend the manuscript for printing process.

Author Response

Dear reviewer,

Thank you very much for your contribution and suggestions.

Best regards,

Authors.